# The Influence of Oncogenic Viruses in Renal Carcinogenesis: Pros and Cons

**DOI:** 10.3390/pathogens11070757

**Published:** 2022-07-02

**Authors:** Bianca Manole, Costin Damian, Simona-Eliza Giusca, Irina Draga Caruntu, Elena Porumb-Andrese, Catalina Lunca, Olivia Simona Dorneanu, Luminita Smaranda Iancu, Ramona Gabriela Ursu

**Affiliations:** 1Department of Morphofunctional Sciences I-Histolgy, Pathology, “Grigore T. Popa” University of Medicine and Pharmacy, 700115 Iasi, Romania; bianca.manole@umfiasi.ro (B.M.); simona-eliza.giusca@umfiasi.ro (S.-E.G.); irina.caruntu@umfiasi.ro (I.D.C.); 2Department of Microbiology, Faculty of Medicine, “Grigore T. Popa” University of Medicine and Pharmacy, 700115 Iasi, Romania; olivia.dorneanu@umfiasi.ro (O.S.D.); luminita.iancu@umfiasi.ro (L.S.I.); ramona.ursu@umfiasi.ro (R.G.U.); 3Department of Dermatology, Faculty of Medicine, “Grigore T. Popa” University of Medicine and Pharmacy, 700115 Iasi, Romania; elena.andrese1@umfiasi.ro

**Keywords:** renal cell carcinoma, urothelial carcinoma, oncogenic viruses, biomarkers, oncolytic viruses

## Abstract

Viral infections are major contributors to the global cancer burden. Recent advances have revealed that known oncogenic viruses promote carcinogenesis through shared host cell targets and pathways. The aim of this review is to point out the connection between several oncogenic viruses from the *Polyomaviridae*, *Herpesviridae* and *Flaviviridae* families and renal carcinogenesis, highlighting their involvement in the carcinogenic mechanism. We performed a systematic search of the PubMed and EMBASE databases, which was carried out for all the published studies on RCC in the last 10 years, using the following search algorithm: renal cell carcinoma (RCC) and urothelial carcinoma, and oncogenic viruses (BKPyV, EBV, HCV, HPV and Kaposi Sarcoma Virus), RCC and biomarkers, immunohistochemistry (IHC). Our analysis included studies that were published in English from the 1st of January 2012 to the 1st of May 2022 and that described and analyzed the assays used for the detection of oncogenic viruses in RCC and urothelial carcinoma. The virus most frequently associated with RCC was BKPyV. This review of the literature will help to understand the pathogenic mechanism of the main type of renal malignancy and whether the viral etiology can be confirmed, at a minimum, as a co-factor. In consequence, these data can contribute to the development of new therapeutic strategies. A virus-induced tumor could be efficiently prevented by vaccination or treatment with oncolytic viral therapy and/or by targeted therapy.

## 1. Introduction

For kidney cancer (C64–C65), the estimated number of new cases in Romania for 2020, for both genders and all ages, was 2750, with a 7.7 age-standardized rate, placing Romania in the second half of incidence rankings for all the European countries [1]. Renal cell carcinoma (RCC) is the most common form of kidney cancer, with around 90 percent of all kidney cancers being attributed to RCC. The grading system for RCC was reviewed and validated recently by the International Society of Urological Pathology (ISUP) and the World Health Organization (WHO) [2,3]. The Genitourinary Pathology Society (GUPS) reviewed recent advances in renal neoplasia, particularly after WHO’s 2016 classification, to provide an update on existing entities, including diagnostic criteria, molecular correlations and updated nomenclature. Key prognostic features for RCC remain WHO/ISUP grade, pTNM stage, coagulative necrosis, and rhabdoid and sarcomatoid differentiation [4].

Tumor progression frequently affects normal renal status, and in evolution the patients may reach renal kidney failure. Onconephrology recognizes the further factors involved in end-stage renal disease patients with cancer: acquired cystic disease of the kidney, cyclophosphamide use in patients with systemic autoimmune diseases, oncogenic viral infections and prolonged use of analgesics, which demonstrates the importance of a multidisciplinary approach to kidney cancer. Screening of renal viral-induced cancers could lead to optimizing the treatment of patients with chronic kidney disease [5].

As part of tumor pathology, gaining knowledge of kidney tumors’ development and the genic and molecular sequences involved in the carcinogenic mechanism, represents an important issue with a direct impact on scientific achievements (contributing to the current state of the art in carcinogenesis), technological advances (by developing and/or improving tools for performant investigations) and socio-economics (mirrored by a possible reduction in the global burden of this malignancy in the general population through better monitoring and new targeted treatment options). Within this framework, the involvement of oncogenic viruses represents a hot topic in the inter-disciplinary approach to renal carcinogenesis.

A growing list of viral oncogenic mechanisms has now been established, including inhibition of apoptosis, reprogramming of host metabolism, modulation of the cellular microenvironment, attenuation of host immune control, transcriptional reprogramming and epigenomic reprogramming [6].

RCC findings have been completed with data regarding the involvement of oncogenic viruses in bladder urothelial neoplasia which have been recently classified by WHO and explored with regard to molecular evolution and classification for the benefit of patient management [7].

Therefore, this paper aims to analyze the role of oncogenic viruses in the initiation and progression of kidney tumors, focusing on the linkage of the specific molecular profile of viruses with tumor behavior in terms of aggressiveness. The published evidence on this topic is not sufficient to support a systematic review; therefore, data are summarized in a narrative, descriptive manner. 

The oncogenic role of some viruses in different types of tumors has been well established by the International Agency for Research on Cancer (IARC), the following examples being relevant: Hepatitis B Virus (HBV) and Hepatitis C Virus (HCV) are involved in liver carcinogenesis; Epstein–Barr Virus (EBV) is known to be responsible for Burkitt lymphoma in parallel with some co-factors; high-risk types of Human Papilloma Virus (HPV) are associated with cervical cancer and oropharyngeal cancers [8]. Recent scientific data support the oncogenic role of some viruses in the carcinogenesis of RCC and urothelial carcinoma (UCa). From among these oncogenic viruses (polyomaviruses, EBV, HCV, HPV, Human Immunodeficiency Virus (HIV)) recognized by IARC, we will refer here specifically to BK Polyoma Virus (BKPyV), EBV and HCV.

## 2. Renal Cell Carcinoma: Carcinogenesis Mechanisms of Oncogenic Viruses

### 2.1. Polyomaviridae Family

*Polyomaviridae* is a family of small, icosahedral, non-enveloped viruses, with a double-stranded DNA genome, 5 kbp in length, the DNA packed together with histones belonging to the host cells. These viruses infect species of mammals, birds and fish [9]. It is considered one family which probably has oncogenic members that could be causative in the development of urothelial tumors. The number of known human polyomaviruses (PyVs) has increased rapidly, from two species—JC Polyomavirus (JCPyV) and BKPyV, detected in 1971—to 13–14 species disclosed since 2007. Most of these viruses were isolated from nasal, skin, serum or stool samples. Despite their recent discovery, many human PyVs are common in the population, and several, including Merkel Cell Carcinoma Polyomavirus (MCPyV) and human PyVs 6 and 7, are frequently found in skin samples [10].

Polyomaviruses are also analyzed in association with some cancers: MCPyV was first discovered clonally integrated in Merkel cell carcinoma [9], and the primary site of productive BKPyV virus replication is the urothelium covering the human urinary tract [11].

Polyomaviruses infect mammal host cells via different glycan receptors, with each virus’ capsomers having an affinity with specific branching glycans. The presence or absence of these receptors on different cellular types can partly indicate the specific cellular tropism of each human polyoma virus, for example, MCPyV in skin cells and BKPyV and JCPyV in the epithelium of the gastrointestinal and genitourinary tract [12]. Once the cytosol has been reached, viral decapsidation is initiated and the following steps of replication and assembly take place in the cell nucleus. A precursor mRNA molecule is then spliced to produce large-T (LT-ag) and small-T (ST-ag) antigens, which are early proteins important in regulating further replication steps. Late transcripts are translated into the capsid proteins VP1, VP2 and VP3. After assembly, virions can be released from host cells both lytically and non-lytically [13]. In permissive cells, the lytic infection model usually occurs, leading to the destruction of the host cell. Viral oncogenesis has been linked to infection occurring in nonpermissive cells in which an abortive infection occurs, with the gene expression being stuck in the LT-ag expression step. The accumulation of this protein deregulates the cell cycle and is considered a key element in the process of carcinogenesis [14].

In a recent retrospective study (2021) of a single-center cohort of kidney transplant recipients (KTRs), 20% of patients were positive for BKPyV before clear cell RCC (ccRCC) development. The aim of this study was to determine the extent of HPyV reactivation in the anatomical sites where these tumors had arisen in order to establish a potential association between ubiquitous virus reactivation in the context of long-lasting iatrogenic immunosuppression and cancer development. Immunohistochemistry (IHC) for the anti-Large T SV40 (clone MRQ-4) and anti-p16INK4a (clone E6H4) was used to confirm the association between BKPyV reactivation and cancer development in kidney transplant recipients (KTRs). These observations strengthen the idea that BKPyV may contribute to malignancies in its respective sites of infection, indicating the need for further investigations into this potential cancer-causing factor in KTRs [15]. Another research team, using nested polymerase chain reaction (PCR), found that 23 (14.3%) samples were positive for BKPyV DNA in a pool of 160 RCCs, bladder UCas and corresponding control samples from patients with benign renal and bladder pathology. In that study, the presence of BKPyV DNA resulted in a fivefold increase in the risk of development of RCC [16]. An interesting study analysed, besides urothelial tumors of the kidneys, five RCC cases for JCPyV/BKPyV infection, and all of these samples (5/5) were JCPyV DNA positive. This study underlines the possible association between polyomaviruses, other than BK, with UCa [17]. The first adult case of RCC associated with BKPyV was documented by means of positive IHC reaction for the SV40 T–Ag (Ab-2) antibody; in situ hybridization (ISH) with a BKPyV probe, which highlighted viral inclusions; and real-time polymerase chain reaction (PCR) analysis with BKPyV primers [18]. A recent study revealed evidence for the potential oncogenic role of BKPyV in collecting duct carcinoma, in renal allografts, by a positive IHC for simian virus 40 (SV40)-LTag and PyVAN-B, which were correlated with a whole-genome sequencing of the tumor which found multiple BKPyV integration sites [19].

The mechanism of multistage BKPyV carcinogenesis in the development of RCC was investigated by a Harvard team of researchers: using next-generation virome capture sequencing, they detected viral integration in BKPyV-associated UCa in renal transplant recipients and concluded that the integration of BKPyV was a continuous process, occurring in both primary and metastatic tumors, generating heterogenous tumoral cell populations [20]. The hypothesis that integration of polyomaviruses is essential to tumorigenesis is sustained by other authors [21,22,23].

#### BKPyV Carcinogenesis

The carcinogenesis of RCC induced by BKPyV has been analysed by many authors [19,20,21,22,23,24,25,26]. The main mechanisms detected were selected here chronologically. The oncogenic potential of BKPyV was identified by high-throughput sequencing of tumor DNA obtained from a sample of UCa arising in a renal allograft and the viral integration into the human genome was detected [21]. Papadimitriou JC et al. mentioned all the possible mechanisms of BKPyV-induced carcinogenesis: evasion of growth suppression (inactivation of both pRB and p53 and inhibition of apoptosis by inactivation of p53), sustaining proliferative signaling (deregulation of multiple crucial signaling pathways for proliferation, including the phosphoinositide-3 kinase–Akt/ protein kinase B pathway), cell death resistance (antiapoptotic effect of SV40 sTag), enabling of replicative immortality (by induction of telomerase activity), induction of angiogenesis (increased vascular endothelial growth factor levels) and activation of tumor invasion and metastases (fibrotic process following inflammatory reactions) [22]. BKPyV was associated with a special urine cytology with a small cell pattern, similar to high-grade squamous intraepithelial lesions (HSILs). Immunosuppression is also involved in the oncogenicity of this virus. LTag is known to inactivate pRb and p53, which leads to uncontrolled cell replication [24]. Peretti A et al. mentioned that the expression of APOBEC3 was detected by IHC analysis in KTRs, and it is believed that APOBEC3 (A3) proteins (a family of ssDNA cytosine deaminases) could have a role in BKPyV mutagenesis in vivo [25]. Recent technology, such as next-generation virome capture sequencing, is helping us to understand viral integration in renal transplant recipients [20]. Furmaga J et al. underline the oncogenic potential of BKPyV due to the possibility of its integrating with the host genome, leading to overexpression of the viral Tag protein [26]. Integration was reported by Meier RPH et al., who performed whole-genome sequencing of a tumor and confirmed multiple genome BKPyV DNA integrations [19]. Viral DNA integration in tumor cells led to persistent upregulation of early gene proteins in the absence of chronic active replication. The authors underline the potential oncogenic role of BKPyV in collecting duct carcinoma in renal allografts [19].

### 2.2. Herpesviridae Family

The *Herpesviridae* family includes numerous species of viruses with a broad range of hosts, ranging from birds to reptiles to mammals, nine of which specifically infect humans. The virions are spherical, enveloped and have an icosahedral capsid, with a size ranging from 150–200 nm in diameter, in which a linear, double-stranded DNA genome is packed, 125–241 kbp in length, comprising 70–170 genes [27]. One important virus associated with cancers from this family is the Epstein–Barr virus, which infects most of the world’s population and is associated with about 1.5% of cancers worldwide. Since the main sites of infection and persistence of this virus are B cells and epithelial cells, EBV-associated cancers include lymphomas (Burkitt’s, Hodgkin’s, immunoblastic, natural killer (NK) cell, etc.), as well as nasopharyngeal and gastric carcinomas [28]. EBV is also associated with RCC, especially the sarcomatoid histological type, which is more aggressive and has a lower therapy response rate [29]. 

Another recognized oncogenic virus in this family is the Kaposi Sarcoma-associated Herpesvirus (KSHV), or Human Herpesvirus-8 (HHV-8), which in immune-deficient patients can cause three different malignancies: Kaposi sarcoma, primary effusion lymphoma and HHV-8-associated multicentric Castleman disease [30]. 

The replicative cycle of *Herpesviridae* family members occurs in both lytic and latent phases. Infection is initiated by the attachment of viral glycoproteins to different cell surface receptors. For EBV, CD21 is the receptor recognized by this virus on the surface of B cells, while ephrin receptor A2 (EphA2) serves to target epithelial cells [31]. In the case of KSHV, multiple binding receptors are recognized, including heparan sulphate, integrins, EphA2 and EphA4 [32]. After cell entry, the capsid uncoats and viral DNA enters the nucleus by a pore. Immediate early gene products have regulatory functions, and the early genes are then expressed to produce the DNA replication complex. A rolling circle mechanism produces DNA concatemers that are cleaved into genome units, packaged into virion proteins encoded by the late genes. The virions exit the cell by either exocytosis or cell-to-cell spread [27].

Regarding EBV, we mention a prospective study in which the authors tested 27 fresh ccRCC tumor tissues by Real-Time PCR (RT-PCR). Viral infections were diagnosed in ten patients (37.0%), including three adenovirus (ADV) cases (11.1%) and eight EBV cases (29.6%). The authors concluded that EBV and ADV infections are common in renal cancer patients and increase the risk of high-grade RCC presence [33]. Another research team tested, by nested PCR, DNA extracted from frozen RCC tissues of nine patients [34]. They realized double testing—PCR and mRNA in situ hybridization of RCC cases (BamHIW antisense probe)—which demonstrated the direct association of EBV with RCC and concluded that the expression of EBV may be involved in the pathogenesis of RCC and nephroblastoma [34]. EBV was confirmed in formalin-fixed paraffin-embedded (FFPE) samples of RCC by EBV ribonucleic acid (EBV-RNA) encoded by in situ hybridization using an EBER probe and a ready-to-use in situ hybridization kit (EBERs are small non-coding RNA fragments isolated from human cells infected by EBV with the potential to be implicated in carcinogenesis and transformation after interaction with different proteins of the host and ribonucleoprotein synthesis) [34]. Considering the EBER positivity of the tumor cells in patients with RCC, these results suggest that EBV may contribute to tumor development as an etiological factor in RCC [34]. EBV infection and its genotypes were investigated in 73 cases of different types of RCC, using PCR for EBV genotyping and EBV-encoded RNA in situ hybridization (EBER-ISH) [35]. The authors found that EBV was present in tumor-infiltrating B lymphocytes of sarcomatoid RCCs [35].

IHC markers are complementary tools beneficial for confirming viral-induced carcinogenesis, highlighting EBV presence in RCC by double testing: nested PCR and IHC assay for expression of p53, p16INK4a, Ki-67 and NF-κB. The NF-κB p65 signaling pathway was found to be statistically significantly associated with EBV DNA in RCC samples [36].

#### 2.2.1. EBV Carcinogenesis

Using ISH and IHC for EBV, Ng KF et al. found that EBV infection is not involved in the carcinogenesis of squamous cell carcinoma of the upper urinary tract [37], and, in 2006, EBV was considered not to be involved in RCC oncogenesis. Six years later, Betge J et al. analysed EBV presence in patients with gastric cancer and concomitant RCC and concluded that EBV could not be considered as a risk factor for either of the simultaneous viruses studied [38]. 

Due to technological developments, in 2019, EBV genes were found to activate oncogenes, such as Bcl-2 and MYC, as well as signaling pathways, such as NF-κB, JNK, JAK/STAT and PI3K/Akt, and inhibit the tumor suppressors DOK1, PKR, p53, PRDM1, DICE1, PTEN and p27kip1, p21WAF1/CIP1. These genes could be considered future precise molecular targets which could provide an effective solution for the prevention and treatment of EBV-related neoplasms [39].

Chakravorty S et al. identified the oncogenic mechanisms mediated by EBV: higher viral RNA load associated with mutation in cancer driver genes, integration into the host genome, host gene mutations and transcriptional responses. This study detected novel points of interaction between EBV and the human genome and identified novel regulatory nodes and druggable targets for individualized EBV and cancer-specific therapies [40]. 

It is known that single miRNA can function as a tumor suppressor or as an oncogene (oncomiR), while it may also have a dual function. For EBV-related nasopharyngeal carcinoma, it was found that miR-18a is overexpressed and positively correlated with tumor size and TNM stage; that it can influence cell survival, epithelial-to-mesenchymal transition and invasion; and that it is involved in the early stages of tumorigenesis. The activation of miRNA is favoured by latent membrane protein 1 (LMP1) encoded by EBV [41]. In 2002, Wen Y et al. reported that 29.6% of mRNA in analysed RCC samples was EBV-positive as determined by ISH and indirect immunofluorescence staining. According to these authors, the oncogenic mechanisms can be the reactivation of latent EBV and coinfections with other components of the microbiome and EBV in EBV-driven cancers [42].

#### 2.2.2. KSHV Carcinogenesis in RCC

Regarding its tumorigenesis, classic KSHV is supposed to transition from infection to cancer by a lytic or latent pattern of infection of host cells. A very important role is maintained by Kaposi’s sarcoma spindle cells, which develop from mesenchymal stem cells [43]. KSHV was associated with increased risk of cancer in kidney transplant recipients, which led to the screening of patients before transplantation [44].

### 2.3. Flaviviridae Family

The *Flaviviridae* family is comprised of four different viral genera: *Flavivirus*, arthropod-borne viruses which infect many mammal and bird species; *Pestivirus*, infecting pigs and ruminants, which are transmitted by contact with secretions; *Pegivirus*, whose species cause persistent infections in mammals but which have not yet been clearly linked to disease; and *Hepacivirus*, with human hepatitis C virus being a major human pathogen, although related viruses also produce liver infections in other mammals. The virions are typically spherical, enveloped, comprised of one basic capsid protein and two to three envelope glycoproteins, with a size between 40–60 nm and containing a positive-sense, non-segmented RNA genome that is 9–13 kbp in length [45]. While HCV is definitely recognized as an oncogenic virus related to hepatocarcinoma, different research teams have also studied its association with other types of cancer, including renal cell carcinoma, prostate and bladder cancers. In a recent meta-analysis, Ma et al. found that HCV infection is significantly associated with an increased risk of RCC, while for bladder and prostate cancers the study did not find any statistically significant associations [46].

The HCV replication cycle begins with a complex attachment and cell entry process that requires at least six different cell surface proteins. Heparan sulphate and low-density lipoprotein receptors are involved in the attachment process, while the true viral receptors are considered to be CD81 and Scavenger Receptor class B member 1 (SR-B1). The virus is then taken up into the cell by clathrin-mediated endocytosis. In the cytoplasm, the virus becomes uncoated, and replication occurs at the endoplasmic reticulum. The positive sense RNA acts as a template for both viral protein and antigenome RNA synthesis, which will later be involved in genome replication. Viral proteins are synthesized as precursors that are later cleaved by viral and cellular proteases. Virion maturation takes place in the Golgi complex, and mature virions are released either by the very low-density lipoprotein pathway or by membrane vesicles [47].

HCV has been implicated in RCC because of its association with chronic kidney disease. A recent Brazilian study tested RCC patients older than 18 years of age for HCV serology. The authors found a 4.1% prevalence of HCV infection in RCC patients, which was threefold higher than in the general Brazilian population [48]. A systematic review that aimed to investigate the association between HCV infection and the risk of RCC supported the potential of this association [49]. Other papers have presented evidence of a relationship between HCV infection and an increased risk for developing RCC based on the follow-up of a cohort of 67,063 patients HCV-tested between 1997 and 2006 [50] and on a prospective study that compared a cohort of patients with chronic HCV infection and extrahepatic malignancies, including RCC, with a control cohort [51].

A nationwide register-based cohort study in Sweden aimed to evaluate the risk of kidney cancer in relation to HCV infection. The study included a total of 43,000 individuals with chronic HCV infection who were followed-up for 9.3 years, with observation of kidney cancer incidence and mortality. The authors found an association between HCV infection and chronic kidney disease, particularly among women. Contrary to the results mentioned above, they concluded that it is premature to consider HCV infection a risk factor for kidney cancer [52].

Despite all the published results, the mechanism by which HCV could promote RCC is not entirely understood. Three main hypotheses have been put forth to explain the possible association between HCV and RCC. One hypothesis implicates a role for HCV in RCC pathogenesis via the NY-REN-54 protein, an altered ubiquitin-related protein that may promote oncogenesis [53]. A second postulated mechanism is HCV-induced increased expression of serine protease inhibitor Kazal (SPIK)—a protein that inhibits serine protease-related apoptosis with the potential to be involved in different cancers [53]. The third hypothesis implicates HCV inhibition of cytotoxic T cell-dependent apoptosis, which disturbs host immunity and normal tissues and leads to renal oncogenesis [53].

#### HCV Carcinogenesis

In 2011, Wiwanitkit V et al. evaluated the cause–outcome relationship between HCV infection and RCC and found one common protein—NY-REN-54—that might be responsible for the cause–outcome relationship observed between HCV infection and RCC [54]. In a prospective study, Gonzalez HC et al. found that chronic infection with HCV is associated with RCC, underlining the relationship between HCV and extrahepatic malignancies [51]. In 2020, Tsimafeyeu I et al. evaluated the efficacy and safety of nivolumab treatment in metastatic ccRCC patients with or without chronic HCV infection and the authors found that the administration of nivolumab warrants further investigation due to the efficacy and safety profiles observed [55].

### 2.4. Papillomaviridae Family

The *Papillomaviridae* family includes many small, icosahedral, non-enveloped viruses with double-stranded DNA genomes. These viruses show great diversity, infecting a wide range of hosts: mammals, birds, reptiles and fish. Some HPVs (called mucosal, high-risk or alpha HPVs) types are associated with several types of cancers, most importantly with cervical but also with vulvar, penile, vaginal and oropharyngeal carcinomas [56].

Papillomaviruses replicate in epithelial cells. After microabrasion, the viruses interact with heparan sulphate, which triggers capsid conformation changes that allow the viruses to be transferred to an as yet-unknown entry receptor. After internalization by a process that resembles macropinocytosis, the DNA remains attached to the capsid protein L2, which facilitates transport to the trans-Golgi network. During the metaphase, the viral DNA becomes associated with host chromosomes. Initially, viral replication produces a low number of copies which is maintained constantly in proliferating cells. After cell differentiation, replication begins generating virions, amplifying viral DNA to high copy numbers, and producing capsids that pack the genome. As the top layers of epitheliums are shed off, the virions are released [57]. 

#### HPV Carcinogenesis

In 1996, Pelisson I et al. detected an abnormal distribution of HPV types in skin squamous cell carcinoma from KTRs and alterations of c-myc, c-Ha-ras and p53 genes, regardless of HPV type. The authors considered that viral infection and oncogene activation could represent factors involved in the etiology of skin squamous cell carcinoma [58]. Dogra S et al. demonstrated that HPV16 E6/E7 altered DNA damage response through p53-mediated cell growth controls, which could be relevant in treating RCC [59].

## 3. Discussion

### 3.1. Towards a Standardized Approach for Testing Oncogenic Viruses in Kidney and Bladder Malignancies

The causal relationship between RCC and/or UCa and oncogenic viruses is controversial, with different results published for each analyzed virus.

The viral hepatitis viruses (HCV and HBV) were detected mainly as single-case presentations in RCC patients, as simple associations. The number of viral-induced RCCs increased in the case of EBV and it was highest in the case of BKPyV. Especially for BKPyV, mechanisms of viral integration, carcinogenesis and reactivation were identified as proof of the role of this virus in RCC carcinogenesis. The literature cited in Table 1, Table 2, Table 3 and Table 4 contains the results of studies realized for a single oncogenic virus tested by PCR and/or by IHC analysis. This proves the difficulty of the research on this issue focusing on the possible involvement of viruses in the renal carcinogenic mechanism and the limitations of current approaches.

The viral detection assays are very heterogenous, varying from antigenic detection to histological analysis, from PCR sequencing to antibody detection (Table 1, Table 2, Table 3 and Table 4). For scientific comparisons between different patients from different countries to be possible, it would be ideal to use a similar screening approach and laboratory assays which have similar optimal sensitivities, specificities and positive and negative predictive values. To this end, we have compiled a list of different viral markers that can be assayed for each type of oncogenic virus (Table 5). The implementation of clinically validated screening assays at national levels would also be useful. Further studies are needed to assess the relevance of these findings with an increased number of RCC cases.

The review of the literature reveals that, besides the specific viral IHC markers mentioned in Table 5, a large framework of cellular proteins has also been investigated in relation to viral-associated renal and bladder tumor proliferation. These proteins, derived from different cellular processes and not from viral genomes, are useful in confirming pathological diagnoses and assessing patient prognostics.

Thus, in BKPyV-associated malignancies, the positive immunoreaction for PAX8, pan-cytokeratins, CK7 and vimentin and the negative immunoreaction for EMA, CK20, GATA3, p504S, S100, HMB45 and CD45 certify the renal or urothelial cell origin of the tumor [18,24,80]; supplementarily, the positive expression of E-cadherin, INI1 and CA9 differentiates a rare variant of RCC, namely, collecting duct carcinoma [80].

The analysis of the p53 tumor protein, which functions as a tumor suppressor, with a role in apoptosis, genomic stability and anti-angiogenesis, indicates a correlation between its positive expression, poor prognosis and advanced tumor clinicopathological features [24,36,65,81,82]. However, p53 mutations appear to be rare in RCC and its specificity is low in this context because its positivity does not necessarily translate into gene mutation [83].

p53 was also studied together with p21, another protein that regulates the cell cycle and apoptosis, in HCV-associated renal tumors [81] in order to explain the relationship with the viral proteins NS5A and NS3 in processes such as the production of reactive oxygen species (ROS) and the inhibition of apoptosis. This study showed that NS3 viral proteins react with the p53 host protein to form a complex that leads to inhibition of p53 function [81]. HCV NS5A also reacts directly with and co-localizes with the p53 protein in the perinuclear region and inhibits p21WAF1 tumor suppressor transcription in a p53-dependent manner [81].

In BKPyV-, EBV- and HCV-associated malignancies, some authors have focused on the p16 protein, which inhibits the normal cell cycle, in order to evaluate the correlation of this biological marker with classical clinicopathological parameters (tumor stage, tumor grade, disease progression) [24,36,79]. The results suggest that p16 could be a potential prognostic marker, useful in predicting the progression of these tumors in viral and non-viral contexts.

Nuclear factor-kappa B (NF-κB) is supposed to be involved in RCC development, its expression being correlated with tumor grade. In EBV-associated RCC, NF-kb nuclear immunopositivity, higher than in non-viral RCC, could sustain EBV interference in the carcinogenesis mechanism by the activation of the NF-κB p65 signaling pathway, leading to the acceleration of tumor progression [36].

In order to evaluate proliferative tumor potential, two markers, namely, proliferating cell nuclear antigen (PCNA) and Ki-67 were studied in HPV- and EBV-associated malignancies, the results revealing their usefulness in estimating tumor aggressiveness [36,82].

The relationship with EVB was confirmed only in the sarcomatoid type of RCC [35]. In this aggressive form of RCC, analysis of intratumoral lymphocyte infiltrate (TIL) showed that it is predominantly constituted by B lymphocytes that show plasma cell differentiation, in which EBV was located exclusively [35]. This fact was demonstrated by the strong immunoreactivity for both CD79a and VS38 (markers for B lymphocytes and plasma cells, respectively) and immunonegativity for CD3 (a marker for T lymphocytes), CD68 (a marker for macrophages) and CD56 (a marker for NK cells) [35].

The immune infiltrate was also investigated in a rare case of Kaposi sarcoma developed in a renal allograft ureter, where CD34, together with HHV8, defined the diagnosis [83]. Selected additional markers—CD8 (a marker for T suppressor lymphocytes), CD19 (a marker for B lymphocytes) and CD69 (a marker for the differentiation of regulatory T (Treg) cells)—were assessed to monitor reduced immunosuppression by discontinuing MMF and gradually reducing CyA [83].

### 3.2. The Benefit of Oncolytic Viral Therapy

Oncolytic viral therapy has become a promising approach for tumor immunotherapy for many types of solid tumors. The most frequently used oncolytic viruses include adenovirus, herpes simplex virus, Newcastle disease virus, measles virus, reovirus and parvovirus. These viruses, which have been studied in many research settings, are known to have minimal human toxicity and to destroy tumoral cells specifically and they have a wide spectrum of anticancer activity. In a systematic review and meta-analysis, the authors studied 4000 patients treated for different cancers with oncolytic viruses (OVs) and concluded that this therapy was effective and well-tolerated as a treatment for advanced solid cancers and that it represents a promising therapeutic approach for solid cancers [84].

There are some clinical trials studying the effects of different OVs in different cancers: a modified herpes simplex virus-1 oncolytic virus OrienX010 in patients with melanoma [85]; Enadenotucirev, a tumor-selective and blood stable adenoviral vector in platinum-resistant ovarian cancer [86]; the oncolytic intratumoral activity of Coxsackievirus A21 (V937) in patients with melanoma [87]; and OBP-301 (Telomelysin), an attenuated type-5 adenovirus that contains the human telomerase reverse transcriptase promoter to regulate viral replication in oesophageal cancer patients [88].

In the last 5 years, there have been several studies [89,90,91,92] which have presented their results after using different OVs in treating RCC. Zhang C et al., combined an oncolytic adenovirus carrying decorin with a CAR-T targeting carbonic anhydrase IX (CAIX) for renal cancer cell therapy. This therapeutical association was evaluated as efficient, as the authors recorded tumor regression and activation of the inflammatory immune system [89]. A multidisciplinary team from Finland designed an oncolytic adenovirus that secretes a cross-hybrid Fc-fusion peptide against PD-L1 which is able to elicit effector mechanisms of an IgG1 as well as IgA1 and consequently activate neutrophils. This therapy with optimised tumor killing efficacy was achieved by means of the expression of an oncolytic adenovirus with limited toxicity [90]. Wang H et al. studied the oncolytic potential of bluetongue virus (BTV), the prototype virus in the genus Orbivirus, within the family Reoviridae, in in vitro and in vivo experimental models of human renal cancer. The authors found that BTV has an oncolytic potential in experimental models of human renal cancer and can trigger apoptosis in RCC cells via a mitochondria-mediated pathway [91]. We identified a clinical trial which evaluated Enadenotucirev (formerly ColoAd1), a tumor-selective chimeric adenovirus, in therapy for RCC patients. The authors concluded that their study provided data to support the use of IV infusion of enadenotucirev, which has the potential to stimulate the immune system [92].

## 4. Conclusions

The study of viral-induced RCC could lead to the development of new therapeutic strategies. A virus-induced tumor could be prevented efficiently by vaccination or be treated by oncolytic viral therapy and/or by targeted therapy, in parallel with preventive measures for other co-factors.

Therefore, a future research direction could develop a comprehensive and integrated analysis of at least three oncogenic viruses (e.g., BKPyV, EBV and HCV) to confirm their role in kidney carcinogenesis. For this purpose, a screening platform for oncogenic virus detection would be useful based on PCR and IHC assays.

## Figures and Tables

**Table 1 pathogens-11-00757-t001:** BKPyV association with RCC and bladder cancer.

Authors, Year, Country	Sample Type	BKPyV Detection Assay/Status	Results	Novelty
Loria SJ et al., 2022 New York, NY, USA [60]	Cadaveric renal transplant	Persistent BKPyV viruria	Invasive small cell bladder carcinoma, with prominent adenocarcinoma component	Molecular evidence of BKPyV DNA in the bladder cancer cells
Chen JM et al., 2021 New York, NY, USA [61]	Kidney transplantation	High-grade UCa	Tumor had metastasized to one left obturator lymph node but spared the two native kidneys and ureters	- BKPyV infection, prolonged post-transplantation history; - De novo tumorigenesis of the graft kidney
Meier RPH et al., 2021 Geneva, Switzerland [19]	Kidney–pancreas female recipient with a history of BKPyV nephritis	Whole-genome sequencing of the tumor confirmed multiple BKPyV genome integrations; persistently elevated anti-BKPyV IgG titres and a specific anti-BKPyV T cell response	Presence of BKPyV oncogenic large tumor antigen (LT-Ag) was identified in large amount within the kidney tumor	Potential oncogenic role of BKPyV in collecting duct carcinoma in renal allografts
Borgogna C et al., 2021 Novara, Italy [15]	ccRCC and UCa	IHC and fluorescent in situ hybridization (FISH) for detection of BKPyV infection	- LT-Ag labelling of tumoral cells was detected in two out of five UCa; - Proportion of BKPyV DNA-FISH-positive UCs nuclei was much lower than that of LT-positive cells	Highlighting the association between BKPyV reactivation and cancer development in KTRs
Cuenca AG et al., 2020 Boston, USA [62]	BKPyV-associated nephropathy and, 6 years later, locally advanced UCa	Bioptic confirmation of BKPyV nephropathy	BKPyV DNA was detected in urine at values greater than 500 × 10^6^ copies/mL	IHC for BKPyV LT-Ag
Wang Y et al., 2020 Guangzhou, China [20]	BKPyV-associated UCa after renal transplantation	Next-generation virome capture sequencing	332 viral integration sites were identified in the six tumors	Integration of BKPyV was a continuous process occurring in both primary and metastatic tumors, generating heterogenous tumor cell populations
Querido S et al., 2020 Lisbon, Portugal [63]	KTR who developed a high-grade UCa 5 years after a diagnosis of JCV nephropathy and 9 years after kidney transplantation	Neoplastic tissue was positive for JCPyV DNA by RT-PCR	IHC staining showed strong positivity for early viral protein JCV LT-ag, using a broad polyomavirus antibody	The first report of high-grade UCa associated with JCPyV nephropathy in a KTR
Chu YH et al., 2020 Madison, WI, USA [64]	Post-transplantation UCa and RCC	LT-Ag-expressing UCas were high-grade, with p16 and p53 overexpression	Tumor genome sequencing revealed BKPyV integration	Post-renal transplantation BKPyV-associated UCas are aggressive and genetically distinct from most non-BKPyV-related UCas
Singh G et al., 2019 New Delhi, India [65]	Plasma urine kidney biopsy	IHC qPCR	- Plasma BKPyV titers were 2.7 × 10^3^ copies/mL, and urine titers were 3.6 × 10^5^ copies/mL; - The tumor cells showed diffuse and strong nuclear reactivity for SV40 T antigen and p53	Rare scenario of the development of a pelvic BK polyoma virus-associated UCa in the nonfunctioning graft of the recipient of a second kidney
Odetola OE et al., 2018 Maywood, IL, USA [24]	Case 1: serum BKPyV titers	BKPyV viremia recurred and peaked at 1 year post-transplantation	Both on the smears and resection specimens, the neoplastic cells were found to be positive for SV40	- BKPyV-associated urologic malignancies in KTR can have a fatal outcome; - Necessity to identify the presence of BKPyV in urological malignancies diagnosed in KTR with reactivated BKPyV infection
	Case 2: serum BKPyV titers	Bladder barbotage urine specimen showed decoy cells	Tumor cells were positive for SV40	
Fu F et al., 2018 Guangzhou, China [66]	Urine FFPE frozen graft tumor tissue	qPCR IHC Deep sequencing and sequence analysis	Integration of genotype IV BKPyV genome into the non-coding RNA (ncRNA) intronic region of human chromosome 18	BKPyV integrated into human genome at new breakpoints and revealed the potential oncogenic mechanism of BKPyV
Csoma E et al., 2016 Debrecen, Hungary [67]	FFPE renal neoplasms, bladder cancer and kidney biopsy	RT and nested PCR	Malignant renal tumors (0/89); urinary bladder carcinoma (0/76)	There is no evidence that WUPyV, KIPyV or HPyV9 have any role in oncogenesis
Kenan DJ et al., 2015 Chapel Hill, NC, USA [21]	FFPE high-grade UCa arising in a renal allograft	Laser capture microdissection RT-PCR Deep sequencing and sequence analysis	- High-grade UCa deeply invading the renal medulla; - High levels of LT-Ag expression in tumor nuclei; - Genomic DNA sequencing revealed a novel integrated BKPyV	First evidence for a high-grade UCa arising in a renal allograft associated with BKPyV fully integrated into the tumor genome at a single location
Saleeb R et al., 2015 Toronto, Canada [68]	High-grade UCa in allograft kidney and bladder	PCR	BKPyV genome present in tumor	A significant proportion of malignancies developed in a renal transplant cohort (4 out of 106 patients, 3.8%)
Bulut Y et al., 2013 Elazig, Turkey [16]	FFPE RCC	Nested PCR for detection of BKPyV DNA and real-time RT-PCR for determining mRNA levels of BKPyV	BKPyV VP1 was present in 69.5% of the BKPyV DNA positive samples	Presence of BKPyV DNA resulted in a fivefold increase in the risk of development of RCC
Neirynck V et al., 2012 Brussels, Belgium [69]	FFPE RCC	IHC of SV40	SV40-positive RCC in allograft	BKPyV plays a role in the occurrence of RCC

**Table 2 pathogens-11-00757-t002:** Herpes virus association with RCC.

Authors Year, Country	Sample Type	Herpes Viruses Detection Assay/Status	Results	Novelty
Farhadi A et al., 2022 Shiraz, Iran [36]	FFPE RCC	Nested PCR for EBV DNA amplification	EBV was found to be significantly associated with RCC	p65 NF-κB signaling pathway is involved in EBV-mediated RCC pathogenesis
Dornieden T et al., 2021 Berlin, Germany [70]	Lymphocytes derived from blood FFPE RCC	Flow cytometry; highly advanced histology (multi-epitope ligand cartography) methods	- EBV, CMV, BKPyV antigen-specific; - CD8^+^ T cells were enriched in the effector memory T cell population in the kidney	Extensive overview of tissue-resident memory T cells’ phenotypes and functions in the human kidney presented for the first time, pointing toward their potential relevance in kidney transplantation
Kryst P et al., 2020 Warsaw, Poland [33]	Partial or radical nephrectomy FFPE RCC	Isolation of the nucleic acids from plasma	Viral infections were diagnosed in ten patients (37.0%): - Three ADV cases (11.1%); - Eight EBV cases (29.6%)	EBV and ADV infections are common in RCC patients and increase the risk of high-grade RCC
Karaarslan S et al., 2018 Izmir, Turkey [71]	FFPE RCC	EBV-encoded early RNA EBER—in situ hybridization EBER probe	- EBER positivity was found in 14/90 RCCs at varying rates; - EBER positivity was also found in renal tubular epithelium in 27/78 cases	EBV may contribute to tumor development as an etiological factor in patients with RCC
Kang MJ et al., 2013 Jeonbuk, Korea [72]	FFPE ccRCC	EBER—in situ hybridization EBER probe	EBV positivity in 67/140 ccRCCs	EBV infection was significantly associated with poor survival of ccRCC patients
Hesser CR et al., 2018 Berkeley, CA, USA [73]	KSHV-positive RCC cell line	Cell culture siRNA experiments RT-qPCR	Methylation at the N6 position of adenosine is centrally involved in regulating KSHV gene expression	KSHV reactivation
Ghaninejad H et al., 2009 Tehran, Iran [74]	Renal transplantation	Dermatological examination	2 cases of Kaposi’s sarcoma	Kaposi’s sarcoma described as most common post-transplant cancer in developing countries

**Table 3 pathogens-11-00757-t003:** Hepatitis virus association with RCC and bladder cancer.

Authors Year, Country	Sample Type	Hepatitis Virus Detection Assay/Status	Results	Novelty
Ma Y et al., 2021 Sichuan, China [46]	FFPE RCC	Positive for anti-HCV and HCV-RNA analysis by RT-PCR	The association of HCV with RCC was most strong (RR = 1.71) in the USA	HCV infection was significantly associated with increased RCC risk
Rangel JCA et al., 2021 Rio de Janeiro, Brazil [48]	FFPE RCC	Antibodies against HCV	4.1% HCV infection from all RCC tested samples	A 3-fold higher prevalence of HCV infection identified among patients with RCC, compared to the general Brazilian population
Liţescu M et al., 2020 Bucharest, Romania [75]	* Primary renal lymphoma	Initiation of direct-acting antiviral therapy	Child–Pugh class A HCV cirrhosis	Discovered incidentally in a patient investigated for HCV
Cormio L et al., 2017 Foggia, Italy [76]	Uca plasmocytoid variant bladder metastasis	74-year-old woman with HCV-related liver cirrhosis	Ascites and no urinary or other symptoms	First reported case of asymptomatic UCa and associated metastasis of hepatocellular carcinoma
Akar E et al., 2019 Istanbul, Turkey [77]	Metastatic RCC	60-year-old man, 16 months after sunitinib initiation	Elevated liver enzymes and hepatitis D virus infection reactivation in the HBsAg-positive patient	Cancer patients should be screened for viral hepatitis prior to immunosuppressive therapy or chemotherapy

***** Although this malignancy is not RCC, the inclusion in the table is sustained by the rarity of the diagnosis.

**Table 4 pathogens-11-00757-t004:** HPV association with RCC.

Authors Year, Country	Sample Type	HPV Detection Assay/Status	Results	Novelty
Henley JK et al., 2017 Danville, PA, USA [78]	Skin biopsy	Enlarged keratinocytes with blue cytoplasm and hypergranulosis characteristic of epidermodysplasia verruciformis (EDV)—features suggestive for HPV infection	Renal transplantation 7 years prior	Rare case of acquired EDV in a solid organ transplant recipient
Farhadi A et al., 2014 Serdang, Malaysia [79]	FFPE RCC	MY/GP+ consensus primers and HPV-16/18 type specific nested PCRs followed by direct sequencing	HPV genome was detected in 37 cases (30.3%) HPV-18 was the most common viral type identified followed by HPV-16 and HPV-58	Results indicate an association of HR-HPV types with RCC

**Table 5 pathogens-11-00757-t005:** Viral and associated IHC markers.

Virus Type	Tumor Type in Renal Allograft	IHC Viral Marker	IHC Associated Markers	References
BKPyV	RCC—collecting duct carcinoma	SV40	PAX8, E-cadherin, CK7, INI1, CA9, vimentin, CK20, GATA3, p504S	Dao M et al., 2018 [80]
	RCC	SV40	Pankeratin, CK7, vimentin, EMA, CK20, S100, HMB45, CD45	Narayanan M et al., 2007 [18]
	RCC	SV40	p53	Singh G et al., 2019 [65]
	Bladder adenocarcinoma UCa	SV40	PAX8, CK7, p53, p16	Odetola OE et al., 2018 [24]
EBV	RCC	EBV LMP-1; EBNA-2; BZLF1	CD79a, CD3, CD68, CD56, CD21, VS38	Kim KH et al., 2005 [35]
	RCC		p53, p16, Ki-67, NF-κB	Farhadi A, 2022 [36]
HCV	RCC	NS3, NS5A	p53, p21	Ahmed et al., 2016 [81]
HPV	RCC	HPLV1 capsid protein	p16	Farhadi A et al., 2014 [79]
	RCC		PCNA, p53 CM-1, p53 DO-7	Kamel D et al., 1994 [82]
HHV-8	Kaposi sarcoma	HHV8	CD34, CD8, CD19, CD69	Dudderidge TJ et al., 2007 [83]

## Data Availability

Data are contained within the article.

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
