# Peer review of "The Influence of Oncogenic Viruses in Renal Carcinogenesis: Pros and Cons"

_pathogens, 2022, doi:10.3390/pathogens11070757_

Round 1

Reviewer 1 Report

Although the paper by Manole et al. sought to investigate a highly relevant and current theme, with high translational potential, some issues must be solved before it can be accepted and published. 

Despite the title suggests that the article sought to investigate the association between oncogenic viruses and RCC, the authors have included in several secondary points that goes far beyond the scope of the review and of the Pathogens journal, such as the discussion about IHC markers, targeted therapy and ctDNA in RCC, which make the article diffuse, hard to follow and superficial in the individual themes discussed.

It would be better to focus on the association between viruses and RCC deepening the discussion around this theme, trying to answer questions like those related to the mechanistic contribution of different viruses to the development of RCC (oncogenic proteins, chronic inflammation, etc.), the prognostic impact of viral infections in RCC, etc.

In this sense, the article sections could be restructured to segregate the discussion for individual viruses, and discuss about the other themes only when they are strictly related to the central theme of the manuscript (e.g.; IHC markers associated to viral infections in RCC, use of viral infections as biomarkers and targets for targeted therapy, etc.).

Also, a brief introduction about the viruses biology (types, subtypes, infectious cycle, cancers associated with them, etc.) would be useful to facilitate the readers comprehension. This is especially important when talking about related viruses, such as different polyomaviruses, since without any introduction about these viruses, it is difficult to interpret the data presented in the text.

Furthermore, the way the article is written seems more a narrative review than a systematic one. Therefore, it would not be a problem to broad the search terms to include other viruses beyond those three included in the paper and to include papers published before 2017. These would increase the volume of relevant data from literature allowing a broaden and deepened discussion.

Author Response

Point-by-point response to the reviewers’ and editor’s comments

Title: “Is renal cell carcinoma induced by oncogenic viruses?”

We thank the reviewer for giving us the opportunity to improve the quality of our manuscript.

Reviewer #1:

Comments and Suggestions for Authors

  • Although the paper by Manole et al. sought to investigate a highly relevant and current theme, with high translational potential, some issues must be solved before it can be accepted and published.
    • Response to Reviewer: We thank the reviewer for this comment. We have tried to optimize then quality of our manuscript thanks to all your suggestions.

  • Despite the title suggests that the article sought to investigate the association between oncogenic viruses and RCC, the authors have included in several secondary points that goes far beyond the scope of the review and of the Pathogens journal, such as the discussion about IHC markers, targeted therapy and ctDNA in RCC, which make the article diffuse, hard to follow and superficial in the individual themes discussed.
    • Response to Reviewer: We thank the reviewer for this comment. We have organized the data as suggested here and bellow. We removed the paragraphs about targeted therapy and about ctDNA in RCC.
  • It would be better to focus on the association between viruses and RCC deepening the discussion around this theme, trying to answer questions like those related to the mechanistic contribution of different viruses to the development of RCC (oncogenic proteins, chronic inflammation, etc.), the prognostic impact of viral infections in RCC, etc.
    • Response to Reviewer: We thank the reviewer for this comment. We have included data regarding oncogenic proteins, chronic inflammation, the prognostic impact of each viral infections in RCC, as suggested.

Lines 163 – 191:

BKPyV carcinogenesis

The carcinogenesis of RCC induced by BKPyV was analysed by many authors. The main mechanisms detected were selected here, chronologically. The oncogenic poten-tial of BKPyV was identified by high-throughput sequencing of tumour DNA obtained from a sample of urothelial carcinoma arising in a renal allograft, and the viral integra-tion into the human genome was detected [21]. Papadimitriou JC et al., mentioned all the possible  mechanisms of  BKPyV-induced carcinogenesis: evasion of growth sup-pression (inactivation of both pRB and p53 and inhibition of apoptosis by inactivation of p53), sustaining proliferative signaling (deregulation of multiple crucial signaling pathways for proliferation, including the phosphoinositide-3 kinase–Akt/ protein ki-nase B), resisting cell death (antiapoptotic effect of SV40 sTag), enabling replicative immortality (by induction of telomerase activity), inducing angiogenesis (increased vascular endothelial growth factor levels), and activating tumor invasion and metasta-ses (fibrotic process following inflammatory reactions) [22]. BKPyV was associated with a special urine cytology with a small cell pattern, similar with High Grade Squa-mous Intraepithelial Lesions (HSIL). Immunosuppression is also involved in onco-genicity of this virus. LTag is known to inactivate pRb and p53 which lead to uncon-trolled cell replications [35]. Peretti A et al., mentioned that the expression of APO-BEC3 by IHC analysis was detected in KTRs, and it is believed that APOBEC3 (A3) pro-teins (a family of ssDNA cytosine deaminases) could have a role in BKV mutagenesis in vivo [40]. The recent technology like next-generation virome capture sequencing is helping us to understand the viral integration in renal transplant recipients [20]. Furmaga J et al., underline the oncogenic potential of BKPyV by its possibility to inte-grate with the host genome and lead to overexpression of the viral Tag protein [41]. Integration was reported by Meier RPH et al., which performed whole genome sequencing of the tumor and confirmed multiple genome BKPyV DNA integrations. Viral DNA integration in the tumor cells led to persistent upregulation of early gene proteins in the absence of chronic active replication. The authors are underlying the potential oncogenic role of BKPyV in collecting duct carcinoma in renal allografts [19].

Lines 246 – 277:

EBV carcinogenesis

Using in situ hybridisation and IHC for EBV, Ng KF et al., found that EBV infection is not involved in the carcinogenesis of SCC of the upper urinary tract [57], and in 2006, EBV was considered not being involved in RCC oncogenesis. 6 years later, Betge J et al., analysed the EBV presence in patients with gastric cancer and concomitant RCC, and they concluded that EBV could not be considered as a risk factor for neither of the simultaneous viruses studied [58].

Due to technology development, in 2019, EBV genes were found to activate onco-genes such as Bcl-2 and MYC, as well as signaling pathways such as NF-κB, JNK, JAK/STAT, and PI3K/Akt, and inhibit tumor suppressor DOK1, PKR, p53, PRDM1, DICE1, PTEN, and p27kip1, p21WAF1/CIP1. These genes could be considered future precise molecular targets which could provide an effective solution for the prevention and treatment of EBV-related neoplasms [59].

Chakravorty S et al., identified the oncogenic mechanisms EBV mediates higher viral RNA load is associated with mutation in cancer driver genes, integration into the host genome, host gene mutations, and transcriptional responses. This study detected novel points of interaction between a EBV and the human genome and identify novel regulatory nodes and druggable targets for individualized EBV and cancer-specific therapies [60].

miR-18a which can function as a tumor suppressor, oncogene(oncomiR), and its function was seen in involvement in early steps of nasopharyngeal carcinoma tumorigenesis. The activation of miRNA is favored by latent membrane protein 1 (LMP1) en-coded by Epstein-Barr virus (EBV) [61].

In 2002, Wen Y et al., mentioned that 29.6% mRNA of analyses RCC samples were EBV positive by in situ hybridization and indirect immunofluorescence staining. According with these authors, the oncogenic mechanisms can be reactivation of latent EBV and coinfections of other components of microbiome and EBV in EBV-driven cancers [62].

KSHV carcinogenesis in RCC

The classical KSHV tumorigenesis suppose transition from infection to cancer, by a lytic or latent pattern of infection of the cells. A very important role is maintained by Kaposi’s sarcoma spindle cells, which develop from mesenchymal stem cells [63]. KSHV was associated with increased risk of cancer in case of kidney transplant recipients, which led to possibility to screen patients before transplantation [64].

Lines 332 – 341:

HCV   carcinogenesis

In 2011, Wiwanitkit V et al, evaluated the cause-outcome relationship between hepatitis C virus infection and renal cell carcinoma, and he found one common protein: NY-REN-54 that there might be cause-outcome relationship between HCV infection and RCC [77]. Gonzalez HC et al., founded in a prospective study that chronic infection with HCV is associated with having RCC, underlying the relationship between HCV and extrahepatic malignancies [71]. In 2020, Tsimafeyeu I et al, evaluated the efficacy and safety of nivolumab in metastatic clear-cell RCC patients with or without chronic HCV infection and the authors found that the administration of nivolumab in warrant further investigation, due to the efficacy and safety profiles observed in this study [78].

Lines 359 – 365:

HPV carcinogenesis

In 1996, Pelisson I et al., detected an abnormal distribution of HPV types in SCC from renal transplant recipients, and alterations of c-myc, c-Ha-ras, and p53 genes without regardless any HPV type. The authors considered that viral infection and oncogene activation could represent factors involved in the etiology of skin SCC from transplant recipients [83]. Dogra S et al., demonstrated that HPV16 E6/E7 altered DNA damage response through p53-mediated cell growth controls, which could be relevant for treating RCC [84].

  • In this sense, the article sections could be restructured to segregate the discussion for individual viruses and discuss about the other themes only when they are strictly related to the central theme of the manuscript (e.g., IHC markers associated to viral infections in RCC, use of viral infections as biomarkers and targets for targeted therapy, etc.).
    • Response to Reviewer: We thank the reviewer for this comment. We have included data regarding IHC markers associated to viral infections in RCC, use of viral infections as biomarkers and targets for targeted therapy, for each virus in part, as suggested.

Lines 154 – 162:

For BKPyV, confirmatory analysis was performed using IHC for SV40 T-Antigen [24, 25]. Like in other cancers [26], in which was possible to differentiate the precan-cerous lessions by invasive cancers, IHC biomarkers can be used to assess progression of RCC: Zhang L et al., analyzed, by IHC, the expression of METTL14 and Pten, and they found that METTL14 might be a potential prognostic biomarker and effective therapeutic target for clear cell RCC [27]. By IHC assay also, the potential role of tu-mor-associated macrophages (TAMs) HIF-1α on (Hippel-Lindau tumor suppressor) was found, in ccRCC progression and supported the role of HIF-1α as a therapeutic target and marker of disease progression [28].

Lines 242 – 245:

IHC markers are complementary tools beneficial to confirm viral-induced carcin-ogenesis, highlighting the EBV presence in RCC by double testing: nested polymerase chain reaction and immunohistochemistry assay for expression of p53, p16INK4a, Ki-67 and NF-κB. NF-κB p65 signaling pathway was found to be statistically significant associated with EBV DNA in RCC samples [51].

Lines 397 – 398:

Table 5: Specific viral IHC markers 

Virus type

IHC Marker

References

BKV

PAX8, E-cadherin, CK7, INI1, focal CA9, Vimentin, SV40

Dao M et al., 2018 [85]

SV40

Narayanan M et al., 2007 [18]

EBV

p53, anti-p16INK4a, ki67, NF-κB (p65

Farhadi A et al., 2022 [51]

CD79a, CD3, CD68, CD56, VS38 – lymphoid marker

EBV LMP-1; EBNA-2, BZLF1 SI CD21 – specific viral marker

Kim KH et al., 2005 [50]

VHC

NY-REN-54

Wiwanitkit V., et al., 2022 [77]

HPV

p16INK4a, HPLV1 capsid protein

Farhadi A et al., 2014 [82]

p53 CM-1, p53 DO-7

Kamel D et al., 1994 [86]

KSV

CD34, HHV8

Dudderidge TJ et al., 2007 [87]

  • Also, a brief introduction about the virus’s biology (types, subtypes, infectious cycle, cancers associated with them, etc.) would be useful to facilitate the readers comprehension. This is especially important when talking about related viruses, such as different polyomaviruses, since without any introduction about these viruses, it is difficult to interpret the data presented in the text.
    • Response to Reviewer: We thank the reviewer for this comment. We have included data regarding types, subtypes, infectious cycle, cancers associated with each analysed virus.

Lines 89 – 153:

2.1. Polyomaviridae is a family of small, icosahedral, non-enveloped viruses, with a double-stranded DNA genome, 5 kbp in length, having the DNA packed together with histones belonging to the host cells. These viruses infect species of mammals, birds, and fish [9].  It is considered one family with probably oncogenic members, that could be causative for the development of urothelial tumors. The number of known human polyomaviruses (PyVs) has increased rapidly, from two species (JCPyV and BKPyV) detected in 1971, to 13-14 species disclosed since 2007. Most of these viruses were isolated from nasal samples, skin, serum, or stool samples. Despite their recent discovery, many human PyVs are common in the population, and several, including MCPyV, and human PyVs 6 and 7 are frequently found skin samples [10].

Polyomaviruses are also analyzed in association with some cancers: Merkel cell polyomavirus (MCPyV) was first discovered clonally integrated in Merkel cell carcinoma [9], and the primary site of productive BKPyV virus replication is the epithelium of the human urinary tract [11].

Polyomaviruses which infect mammal hosts entry cells via different glycan receptors, with each virus’ capsomers having affinity to specific branching glycans. The presence or absence of these receptors on different cellular types can partly indicate the specific cellular tropism of each human PyV, for example MCPyV in skin cells and BKPyV and JCPyV in the epithelium of the gastrointestinal and genitourinary tract [12]. Once reaching the cytosol, viral decapsidation initiates and the following steps of replication and assembly take place in the cell nucleus. A precursor mRNA molecule is then spliced to produce the large-T (LT-ag) and small-T (ST-ag) antigens, which are early proteins important in regulating further replication steps. Late transcripts are translated into the capsid proteins VP1, VP2 and VP3. After assembly, virions can be released from host cells both lytically or non-lytically [13]. In permissive cells, the lytic infection model usually occurs, leading to the destruction of the host cell. Viral onco-genesis has been linked to infection occurring in nonpermissive cells in which an abortive infection occurs, with the gene expression being stuck in the LT-ag expression step. The accumulation of this protein de-regulates the cell cycle and is considered a key element in the process of carcinogenesis [14].

Lines 193 – 241:

2.2. The Herpesviridae family includes numerous species of viruses with a broad range of hosts, ranging from birds, reptiles, and mammals, 9 of which specifically infect humans. The virions are spherical, enveloped, with an icosahedral capsid, having a size ranging from 150-200 nm in diameter, which pack a linear, double stranded DNA genome, 125-241 kbp in length and comprised of between 70-170 genes [42]. One important virus associated with cancers from this family is the Epstein-Barr virus, which infects most of the world population and is associated with about 1.5% of cancers worldwide. Since the main sites of infection and persistence of this virus are B cells and epithelial cells, EBV-associated cancers include lymphomas (Burkitt’s, Hodgkin’s, immunoblastic, NK cell, etc.), as well as nasopharyngeal and gastric carcinomas [43]. EBV is also associated with renal cell carcinoma, especially the sarcomatoid histological type, which is more aggressive and has a lower response rate to therapy [44]. An-other recognized oncogenic virus in this family is the Kaposi Sarcoma-associated Herpesvirus (KSHV), or Human Herpesvirus – 8 (HHV-8), which in the case of immune deficient patients can cause 3 different malignancies: Kaposi sarcoma, primary effusion lymphoma and HHV-8-associated multicentric Castleman disease [45].  

The replicative cycle of the Herpesviridae family members occur in both lytic and latent phases. Infection is initiated by attachment of the viral glycoproteins to different cell surface receptors. For EBV, CD21 is the receptor recognized by this virus on the surface of B cells, while ephrin receptor A2 (EphA2) serves to target epithelial cells [46]. In the case of KSHV, multiple binding receptors are recognized, including heparan sulphate, integrins, EphA2 and EphA4 [47]. After cell entry, the capsid uncoats and viral DNA enters the nucleus by a pore. Immediate early gene products have regulatory functions, and the early genes are then expressed to produce the DNA replication complex.  A rolling circle mechanism produces DNA concatemers that are cleaved into genome units, packaged into virion proteins that are encoded by the late genes. The virions exit the cell by either exocytosis or cell to cell spread [42].

Lines 279 – 305:

2.3. The Flaviviridae family is comprised of four different viral genera: Flavivirus, arthropod-borne viruses which infect many mammal and bird species, Pestivirus, in-fecting pigs and ruminants, and are transmitted by contact with secretions, Pegivirus, which cause persistent infections in mammals, but are not yet clearly linked to diseases, and Hepacivirus, with human hepatitis C virus being a major human pathogen, but also related viruses which produce liver infections in other mammals. The virions are typically spherical, enveloped, comprised of one basic capsid protein and 2-3 envelope glycoproteins, with a size between 40-60 nm, and containing a positive-sense, non-segmented RNA genome that is 9-13 kbp in length [65]. While HCV is definitely recognized as an oncogenic virus related to hepatocarcinoma, different research teams also study its association to other types of cancer, including renal cell carcinoma, prostate or bladder cancers. In a recent meta-analysis, Ma et al. found that HCV infection is significantly associated with an increased risk of RCC, while for bladder and prostate cancers the study did not find a statistically significant association [66].

The HCV replication cycle begins with a complex attachment and cell entry pro-cess, that requires at least 6 different cell surface proteins. Heparan sulphate and low-density lipoproteins (LDL) receptor are involved in the attachment process, while the true viral receptors are considered to be CD81 and Scavenger Receptor class B member 1 (SR-B1).  The virus is then uptaken into the cell by clathrin-mediated endocytosis. In the cytoplasm, the virus becomes uncoated, and replication happens at the endoplasmic reticulum. The positive sense RNA acts as a template for both viral protein and antigenome RNA synthesis, which will later be involved in genome replication. Viral proteins are synthesized as precursors that are later cleaved by viral and cellular proteases. Virion maturation takes place in the Golgi complex and mature virions are re-leased either by VLDL pathway or by membrane vesicles [67]. 

Lines 344 – 358:

2.4. HPV

The Papillomaviridae family includes many small, icosahedral, non-enveloped vi-ruses with double-stranded DNA genome. These viruses show great diversity, infecting a wide range of hosts mammals, birds, reptiles and fish. Some human papillomaviruses (called mucosal, high-risk or alpha HPV) types are associated with several types of cancers, most importantly with cervical, but also with vulvar, penile, vaginal and oropharyngeal carcinomas [79].

Papillomaviruses replicate in epithelial cells. After a micro-abrasion, the viruses interact with heparan sulphate, which triggers capsid conformation changes that allow the viruses to be transferred to a yet-unknown entry receptor. After internalization by a process that resembles macropinocytosis, the DNA remains attached to the capsid protein L2, which facilitates transport to the trans-Golgi network. During the meta-phase, the viral DNA becomes associated with the host chromosomes. Initially, viral replication produces a low number of copies, which is maintained constant in prolifer-ating cells. After cell differentiation, replication begins generating virions, amplifying viral DNA to high copy numbers, and producing capsids that pack the genome. As the top layer of the epitheliums sheds off, the virions are released [80].

  • Furthermore, the way the article is written seems more a narrative review than a systematic one. Therefore, it would not be a problem to broad the search terms to include other viruses beyond those three included in the paper and to include papers published before 2017. These would increase the volume of relevant data from literature allowing a broaden and deepened discussion.
    • Response to Reviewer: We thank the reviewer for this comment. We have extended our research for the last 10 years and we included two new viruses: HPV and Kaposi Sarcoma Virus, as suggested.  

Lines 390 – 395:

Table 1. BKPyV virus association with RCC.

First Author

Year, Country

RCC Sample type

BKPyV detection assay / status

Results

Novelty

Loria SJ et al., 2022

New York, USA

[29]

cadaveric renal transplant

persistent BKPyV viruria

invasive small cell carcinoma adenocarcinoma component

molecular evidence of BKPyV DNA in the cancer cells

Chen JM et al., 2021

NY, USA

[30]

kidney transplantation

high-grade urothelial carcinoma

the tumor had metastasized to one left obturator lymph node but spared the two native kidneys and ureters

BKPyV infection, prolonged post-transplantation history and dissemination

de novo tumorigenesis of the graft kidney

Meier RPH et al., Geneva, Switzerland

[19]

kidney-pancreas female recipient with a history of BKPyV nephritis

 whole genome sequencing of the tumor confirmed multiple genome BKPyV integrations.

-persistently elevated anti-BK virus IgG titres and a specific anti-BKPyV T cell response

the presence of BKPyV oncogenic large tumor antigen (LT-Ag) was identified in large amount within the kidney tumor

potential oncogenic role of BKPyV in collecting duct carcinoma in renal allografts

Borgogna C et al., 2021,

Novara, Italy

[15]

clear cell RCC and urinary bladder carcinomas

immunohistochemistry and fluorescent in situ hybridization (FISH) for detection of BKPyV

 infection

LT-Ag labelling of tumoral cells was detected in two out of five bladder carcinomas.

the proportion of BKPyV DNA-FISH-positive bladder carcinoma nuclei was much lower than that of LT-positive cells

Highlighting the association between BKPyV reactivation and cancer development in KTRs, especially bladder carcinoma

Cuenca AG et al., 2020,

Boston, MA

[31]

BKPyV - associated nephropathy and, 6 years later, locally advanced urothelial malignancy

bioptic confirmation of BKPyV nephropathy

 BKPyV DNA was detected in urine at values greater than 500 x 106 copies/mL

IHC for BKPyV LT-Ag, p53 and Ki-67 positive in atypical cells

Wang Y ET AL., 2020

Guangzhou, China

[20]

BKPyV-associated urothelial carcinoma after renal transplantation

next generation

virome capture

sequencing

332 viral integration sites were identified in the six tumors

integration of BKPyV was a continuous process occurring in both primary and metastatic tumors, generating heterogenous tumor cell populations

Querido S et al., 2020

Lisbon, Portugal.

[32]

KTR who developed a high-grade urothelial carcinoma 5 years after a diagnosis of JCV nephropathy and 9 years after kidney transplantation

Neoplastic tissue was positive for JCPyV DNA by real-time PCR

Immunochemical staining showed strong positivity for the p16, p53, and Ki67 cell cycle markers and for early viral protein JCV LT-ag; using a broad polyomavirus antibody

the first report of urothelial high-grade carcinoma associated with JCPyV nephropathy in a KT recipient

Chu YH et al., 2020

Madison, WI

[33]

post transplantation urothelial carcinomas (UCs) and renal cell carcinomas (RCCs)

LTAg-expressing UCs were high grade with p16 and p53 overexpression

tumor genome sequencing revealed BKPyV integration

post-renal transplantation BKPyV-associated UCs are aggressive and genetically distinct from most non-BKPyV-related UCs

Singh G et al., 2019

New Delhi, India

[34]

plasma

urine

kidney biopsy

IHC

qPCR

Plasma BKPyV virus titers were 2.7 x 103 copies/ml, and urine titers were 3.6 x 105 copies/ml

The tumor cells

showed diffuse and strong nuclear reactivity for simian virus

40 T antigen and p53

rare scenario

of the development of a pelvic BKPyV

associated urothelial carcinoma in the nonfunctioning

graft of the recipient of a second kidney

Odetola OE et al., 2018, Illinois

[35]

Case 1: serum BKPyV virus titers

BK viremia recurred and peaked at 1-year posttransplant

Both on the cell block sections and resection specimens,

the neoplastic cells were found positive for CK7, p16, SV40 (polyomavirus),

p53 and PAX8

BKPyV -associated urologic malignancies

in KT recipients can have a fatal outcome

the need to identify the presence of BKPyV in

urologic malignancies diagnosed in renal transplant recipients with

reactivated BKPyV infection.

Case 2: serum BKPyV virus titers

bladder barbotage urine specimen showed decoy

cells

The tumor cells were

positive for PAX-8, GATA-3 and CK7, and negative for CK20, TTF-1,

CDX2, p63, WT1, D2-40, mammaglobin, ER, S100, PSA and PSAP.

Fu F et al., 2018,

China

[36]

Urine

FFPE

frozen graft tumor tissue 

qPCR

IHC

Deep Sequencing and Sequence Analysis

integration of genotype IV BKPyV genome into the non-coding RNA (ncRNA) intronic region of human chromosome 18

IHC positive of pan

cytokeratins, CK5/6, CK7, CK20, p53, p63, PAX8, and vimentin.

BKPyV integrating into human genome at new breakpoints and

revealed the potential oncogenic mechanism of BKPyV

Csoma E et al., 2016, Debrecen, Hungary

[37]

formalin-fixed paraffin-embedded renal neoplasms, bladder cancer and kidney biopsy

real-time and nested PCR

malignant renal tumours (0/89)

 bladder cancer (0/76)

there is no evidence that WUPyV, KIPyV or HPyV9 have any role in oncogenesis.

Kenan DJ et al., 2015, NC, USA

[21]

FFPE tissue of a high-grade urothelial carcinoma arising in a renal

allograf

Laser capture microdissection

Real-time PCR

Deep sequencing and sequence analysis

high-grade malignant transitional cell carcinoma deeply

invading the renal medulla

high levels

of large T-antigen expression in tumour nuclei

Genomic DNA sequencing revealed a novel integrated BKPyV

the first time that a high-grade

urothelial tumour arising in a renal allograft is associated

with BKPyV fully integrated into the

tumour genome at a single location

Saleeb R et al., 2015, ON, Canada

[38]

High grade papillary urothelial carcinoma in

allograft kidney and bladder, kidney stage pT3,

pNX

PCR

BKPyV viral genome present in tumor

development of malignancies in the allograft kidney was a significant proportion of the malignancies developing in our renal transplant cohort (4 of 106 patients, 3.8%).

Bulut Y et al., 2013, Turkey

[16]

FFPE tissue samples were obtained from patients with RCC

nested PCR for detection of BKPyV DNA and real-time reverse transcription PCR (real-time RT-PCR) for determining mRNA levels of BKV

BKV VP1 was present in 69.5% of the BKPyV DNA positive samples

the presence of BKPyV DNA resulted in a fivefold increase in the risk of development of renal cell carcinoma

Neirynck V et al., 2012, Brussels, Belgium

[39]

FFPE sample

immunohistochemistry of SV40

SV40-positive RCC in her allograft

BKPyV plays a role in the occurrence of this malignant process

Table 2. Herpes viruses association with RCC.

First

Author

Year,

Country

RCC Sample type

Herpes viruses

detection assay / status

Results

Novelty

Farhadi A et al., 2022

Shiraz, Iran

[51]

histologically confirmed FFPE RCC tissue specimens

nested polymerase chain reaction (nPCR) for EBV DNA amplification

EBV was found to be significantly associated with RCC.

p65 NF-κB signaling pathway is involved in EBV-mediated RCC pathogenesis

Dornieden T et al., 2021

Berlin, Germany

[52]

Lymphocytes derived from blood

renal peritumor samples

tumor samples

flow cytometry

highly advanced histology (multi-epitope ligand cartography) methods.

EBV, CMV, BKV antigen specific CD8+ T cells were enriched in the effector memory T cell population in the kidney

extensive overview of TRM cells' phenotypes and functions in the human kidney for the first time, pointing toward their potential relevance in kidney transplantation

Kryst P et al., 2020

Warsaw, Poland

[48]

partial or radical nephrectomy

histologically confirmed RCC

isolation of the nucleic acids NucleoSpin Tissue Kit from tumor tissue and using the EZ1 Virus Mini Kit v2.0 from plasma.

viral infections were diagnosed in ten patients (37.0%): three ADV cases (11.1%) and eight EBV cases (29.6%).

EBV and ADV infections are common in renal cancer patients and increase the risk of high-grade RCC presence.

Karaarslan S et al., 2018

Turkey

[53]

FFPE from patients diagnosed with RCC

EBV-encoded early RNA (EBER) - in situ hybridization

an EBER probe

a ready-to-use in situ

hybridization kit

the EBER positivity was found in 14 of RCCs at varying rates.

The EBER positivity was also found in renal tubular epithelium in 27 / 78 cases.

EBV may contribute to the tumor development as an etiological factor in patients with RCC

Kang MJ et al., 2013

Republic of Korea

[54]

paraffin-embedded tissue

blocks RCC

EBV encoded small RNA (EBER) in situ hybridization using a fluorescein conjugated EBV probe for detection of EBER transcripts

67 / 140 EBV positive

EBV infection was significantly

associated with poor survival of clear cell RCC patients

Hesser CR et al., 2018,

CA, USA

[55]

KSHV-positive renal carcinoma cell line

cell culture

siRNA experiments

RT-qPCR

methylation at the N6 position of adenosine is centrally involved in regulating KSHV gene expression

KSHV reactivation

Ghaninejad H et al, 2009,

Tehran, Iran

[56]

renal transplantation

dermatological

examination

2 cases of Kaposi's sarcoma

Kaposi's sarcoma described as most common post-transplant cancer in developing countries

Table 4. HPV association with RCC.

First Author

Year,

Country

RCC Sample type

HPV detection

assay / status

Results

Novelty

Henley JK et al, 2017,

Pennsylvania, USA

[81]

skin biopsy

enlarged keratinocytes with blue cytoplasm and hypergranulosis characteristic of epidermodysplasia verruciformis

renal transplantation 7 years prior

rare case of acquired EDV in a solid organ transplant recipient.

Farhadi A et al., 2014,

Serdang, Malaysia

[82]

FFPE of confirmed RCC

MY/GP+ consensus primers and HPV-16/18 type specific nested PCRs followed by direct sequencing

HPV genome was detected in 37 (30.3%)

HPV-18 was the most common viral type identified followed by HPV-16 and 58.

the results indicate an association of HR-HPV types with renal cell carcinoma.

Reviewer 2 Report

With below reviews, can we summarize possible mechanisms for immune-competent and compromised host (CKD, ESKD, kidney transplant recipient)

Simona Ruxandra Volovat, Constantin Volovat, Ingrith Miron, Mehmet Kanbay, David Goldsmith, Cristian Lungulescu, Silvia Corina Badarau, Adrian Covic, Oncogenic mechanisms in renal insufficiency, Clinical Kidney Journal, Volume 14, Issue 2, February 2021, Pages 507–515,

Tempera, I. and P. M. Lieberman (2021). "Oncogenic Viruses as Entropic Drivers of Cancer Evolution." Frontiers in Virology

Author Response

Point-by-point response to the reviewers’ and editor’s comments

Title: “Is renal cell carcinoma induced by oncogenic viruses?”

We thank the reviewer for giving us the opportunity to improve the quality of our manuscript.

Reviewer #2:

Comments and Suggestions for Authors

  • With below reviews, can we summarize possible mechanisms for immune-competent and compromised host (CKD, ESKD, kidney transplant recipient)
    • Response to Reviewer: We thank the reviewer for this comment. We have included data from bellow suggested papers in our manuscript:

Lines 46 – 52: Tumor progression frequently affects the normal renal status, and in evolution the patients may reach renal kidney failure. Onconephrology recognize the next factors being involved in end-stage renal disease patients with cancer: acquired cystic disease of the kidney, cyclophosphamide use in patients with systemic autoimmune diseases, oncogenic viral infections, prolonged analgesic, which demonstrates the importance of a multidisciplinary approach of kidney cancer. Screening of renal viral induced cancers could lead to optimizing treatment of patients with chronic kidney disease [5].

Lines 62 – 65: A growing list of viral oncogenic mechanisms is now established, including inhibition of apoptosis, reprogramming host metabolism, modulation of the cellular microenvironment, attenuation of host immune control, transcriptional reprogramming, epigenomic reprogramming [6].

Lines 468 – 471:

  1. Volovat, S.R.; Volovat, C.; Miron, I.; Kanbay, M.; Goldsmith, D.; Lungulescu, C.; Badarau, S.C.; Covic, A. Oncogenic Mechanisms in Renal Insufficiency. Clin Kidney J 2021, 14, 507–515, doi:10.1093/ckj/sfaa122.
  2. Tempera, I.; Lieberman, P.M. Oncogenic Viruses as Entropic Drivers of Cancer Evolution. Front Virol 2021, 1, 753366, doi:10.3389/fviro.2021.753366.

Reviewer 3 Report

In its present form, the extensive collection of literature has some "shotcomings"  which need to be improved.

1. The text is - in some parts - difficult to read because of very long sentence formations. The tables, too, are somewhat overloaded and confusing and should be simplified. A stylistic revision is therefore advisable.

2. Organization of the text

The order in which the 3 most common virus groups (Polyoma viruses [a], Herpesviridae [b], Flaviviridae [c]) are presented changes repeatedly in the text (e.g. line 55-66: c,b,a; line 67-102: b,c,a; line 103-187: a,b,c; and tables c,b,a). A uniform order should be maintained in order to make the review easier to read for "our" readers.

3. Incorrect citation of individual references

In line 105-108 the authors write that Borgogna (14) reports: "20% of the tested clear cell RCC were detected positive for BKPyV by IHC .. and FISH". This is incorrectly quoted. Borgogna states: In his Abstract: "It (the virus) was undetectable in all formalin-fixed and paraffin-embedded blocks obtained from the 10 kidney tumors" and in Results and Discussion: "all the ccRCC resulted negative for LT expression" and later "..even though the virus was no longer detectable in the tumor cells ...". Also, the work of Bulut(15) cited subsequently is not quoted exactly. Bulut does not report, as claimed, on "160 RCC and bladder transitinal cell carcinomas; but in pool of 160  samples there were 50 RCC and 40 transitional cell carcinomas. Improvements are needed in this area.

4. At the 1st citation of a technical term the full name should be written out first immediately followed by the abbreviation in brackets and in the following text only the abbreviation - and not as e.g. in line 24, 61, 63 only the abbreviation or only the written name and only later (line 98) the full name with the corresponding abbreviation; also in lines 209/10 and 249 only abbreviations should be used.

5. line 289: " there are more than 10 studies" - in a review these publications should be included in the references and quoted.

Author Response

Point-by-point response to the reviewers’ and editor’s comments

Title: “Is renal cell carcinoma induced by oncogenic viruses?”

We thank the reviewer for giving us the opportunity to improve the quality of our manuscript.

Reviewer #3:

Comments and Suggestions for Authors

In its present form, the extensive collection of literature has some "shotcomings"  which need to be improved.

  • The text is - in some parts - difficult to read because of very long sentence formations. The tables, too, are somewhat overloaded and confusing and should be simplified. A stylistic revision is therefore advisable.

  • Response to Reviewer: We thank the reviewer for this comment. We have included more data in our tables as suggested by reviewer 1, and we tried to perform a stylistic revision.

  • Organization of the text

The order in which the 3 most common virus groups (Polyoma viruses [a], Herpesviridae [b], Flaviviridae [c]) are presented changes repeatedly in the text (e.g. line 55-66: c,b,a; line 67-102: b,c,a; line 103-187: a,b,c; and tables c,b,a). A uniform order should be maintained in order to make the review easier to read for "our" readers.

  • Response to Reviewer: We thank the reviewer for this comment. We have reorganized our manuscript as suggested: one single paragraph for each analysed virus:

Polyomaviruses: lines 89 - 191

Herpesviruses: lines 193 - 277

HCV: lines 279 - 341

HPV: lines 342 – 365.

  • Incorrect citation of individual references:

In line 105-108 the authors write that Borgogna (14) reports: "20% of the tested clear cell RCC were detected positive for BKPyV by IHC .. and FISH". This is incorrectly quoted. Borgogna states: In his Abstract: "It (the virus) was undetectable in all formalin-fixed and paraffin-embedded blocks obtained from the 10 kidney tumors" and in Results and Discussion: "all the ccRCC resulted negative for LT expression" and later "..even though the virus was no longer detectable in the tumor cells ...". Also, the work of Bulut(15) cited subsequently is not quoted exactly. Bulut does not report, as claimed, on "160 RCC and bladder transitinal cell carcinomas; but in pool of 160 samples there were 50 RCC and 40 transitional cell carcinomas. Improvements are needed in this area

  • Response to Reviewer: We thank the reviewer for this comment. We have checked and made the correction in the text, as suggested.

Lines 121 – 135:

In a recent retrospective study (2021) of a single center cohort of Kidney Trans-plant Recipients (KTRs), 20% of patients were positive for BKPyV, before clear cell RCC (ccRCC) development. The aim of this study was to determine the extent of HPyV reactivation in the anatomical sites where these tumors had arisen, to establish a potential association between the ubiquitous virus reactivation in the context of long-lasting iatrogenic immunosuppression and cancer development. IHC for the anti-Large T SV40 (clone MRQ-4) and antip16INK4a (clone E6H4) was used to confirm the association be-tween BKPyV reactivation and cancer development in KTRs. The observations strengthen the idea that BKPyV may contribute to malignancies in its respective sites of infection, implicating the need for further investigations into this potential cancer-causing factor in KTRs [15]. Another research team detected by nested PCR that 23 (14.3%) samples were positive for BKPyV DNA, in a pool of 160 RCCs, bladder transitional cell carcinomas and corresponding control samples from patients with benign renal and bladder pathology. For that study, the presence of BKPyV DNA resulted in a fivefold increase in the risk of development of RCC [16].

  • At the 1st citation of a technical term the full name should be written out first immediately followed by the abbreviation in brackets and in the following text only the abbreviation - and not as e.g. in line 24, 61, 63 only the abbreviation or only the written name and only later (line 98) the full name with the corresponding abbreviation; also in lines 209/10 and 249 only abbreviations should be used.
    • Response to Reviewer: We thank the reviewer for this comment. We have used used the first name just at the first appearance in the text and we used only abbreviations as suggested. We used only ”BKPyV” in all the manuscript.
  1. line 289: " there are more than 10 studies" - in a review these publications should be included in the references and quoted.
  • Response to Reviewer: We thank the reviewer for this comment. We have modified, as suggested.

Lines 417 – 420:

In the last 5 years, there are several studies [93-96] which are presenting their results after using different OVs in treating RCC. Also, Zhang C et al., combined an oncolytic adenovirus carrying decorin with a CAR-T targeting carbonic anhydrase IX (CAIX) for renal cancer cells therapy.

Round 2

Reviewer 1 Report

Modifications provided by the authors have greatly improved the manuscript quality and significance.

Minnor points to be considered before its publication are detailed bellow:

- In my opinion the stretch between lines 155-162, which discusses IHC markers with no relationship with viral infections, has little to do with the article scope. I would consider to remove it.

- Line 264: The presentation of miR-18a is confusing since it is not clear if it is considered a tumor supressor, a oncomiR or both, depending on the situation. Please clarify.

- In table 5, does the cellular proteins (PAX8, E-cadherin, CK7, p53, Ki67, NF-kB) listed are strong enough to define them as "Specific viral IHC markers", as stated in the table's title? I understand that some of these proteins may be associated with viral infections, however, as proteins derived from cellular processes and not from viral genomes, many of them may be also expressed by virus-negative RCCs. It would be better to separate on the table markers associated with viral infections, ideally discussing their functions in the text, and those viral proteins used to test the presence of viruses in tumor tissue.

Author Response

Point-by-point response to the reviewers’ and editor’s comments

Title: “The influence of oncogenic viruses in renal carcinogenesis: pros and cons”

We thank the reviewer for giving us the opportunity to improve the quality of our manuscript.

Supplementary, we revised tables 1 – 4 and we have corrected all the inaccuracies identified in the first version of the manuscript. Thus, data in tables 1 – 4 are now presented in a more accurate and uniform manner.

Reviewer #1:

Comments and Suggestions for Authors

Modifications provided by the authors have greatly improved the manuscript quality and significance.

Minnor points to be considered before its publication are detailed bellow:

  • In my opinion the stretch between lines 155-162, which discusses IHC markers with no relationship with viral infections, has little to do with the article scope. I would consider to remove it.

o   Response to Reviewer: We thank the reviewer for this comment. We removed the suggested lines 155 – 162.

  • Line 264: The presentation of miR-18a is confusing since it is not clear if it is considered a tumor supressor, a oncomiR or both, depending on the situation. Please clarify.

o   Response to Reviewer: We thank the reviewer for this comment. We clarified the role of miR-18a (lines 253 – 262):

It is known that single miRNA can function as a tumour suppressor, oncogene (oncomiR) or it can have a dual function. For EBV related nasopharyngeal carcinoma, it was found that miR-18a is overexpressed and positively correlated with tumour size and TNM stage, can influence cell survival, epithelial-to-mesenchymal transition and invasion, and is involved in early stages of tumorigenesis. The activation of miR-NA is favoured by latent membrane protein 1 (LMP1) encoded by EBV [41]. In 2002, Wen Y et al., mentioned that 29.6% mRNA of analysed RCC samples were EBV positive by ISH and indirect immunofluorescence staining. According with these authors, the oncogenic mechanisms can be reactivation of latent EBV and coinfections of other components of microbiome and EBV in EBV-driven cancers [42].

  • In table 5, does the cellular proteins (PAX8, E-cadherin, CK7, p53, Ki67, NF-kB) listed are strong enough to define them as "Specific viral IHC markers", as stated in the table's title? I understand that some of these proteins may be associated with viral infections, however, as proteins derived from cellular processes and not from viral genomes, many of them may be also expressed by virus negative RCCs. It would be better to separate on the table markers associated with viral infections, ideally discussing their functions in the text, and those viral proteins used to test the presence of viruses in tumour tissue.

o   Response to Reviewer: We thank the reviewer for this comment. We modified the table as suggested and we mentioned separately the cellular proteins (lines 389 – 440): 

Table 5: Viral and associated IHC markers

Virus type

Tumor type in renal allograft

IHC viral marker

IHC associated markers

References

BKPyV

RCC - collecting duct carcinoma

SV40

PAX8, E-cadherin, CK7, INI1, CA9, vimentin, CK20, GATA3, p504S

Dao M et al., 2018

[80]

RCC

SV40

pankeratin, CK7, vimentin, EMA, CK20, S100, HMB45, CD45

Narayanan M et al., 2007

[18]

RCC

SV40

p53

Singh G et al., 2019

[65]

bladder adenocarcinoma

UCa

SV40

PAX8, CK7, p53, p16

Odetola OE et al., 2018

[24]

EBV

RCC

EBV LMP-1;

EBNA-2;

BZLF1

CD79a, CD3, CD68, CD56, CD21, VS38

Kim KH et al., 2005

[35]

RCC

p53, p16, Ki-67, NF-κB

Farhadi A et al., 2022

[36]

HCV

RCC

NS3, NS5A

p53, p21

Ahmed et al., 2016

[81]

HPV

RCC

HPLV1 capsid protein

p16

Farhadi A et al., 2014

[79]

RCC

PCNA, p53 CM-1, p53 DO-7

Kamel D et al., 1994

[82]

HHV-8

Kaposi sarcoma

HHV8

CD34, CD8, CD19, CD69

Dudderidge TJ et al., 2007

[83]

The review of literature reveals that, beside the specific viral IHC markers mentioned in Table 5, a large framework of cellular proteins is also investigated in the viral-associated renal and bladder tumour proliferation. These proteins, derived from different cellular processes and not from viral genomes, are useful to confirm the pathological diagnosis and to assess the patients’ prognostic.

Thus, in BKPyV - associated malignancies, the positive immunoreaction for PAX8, pan-cytokeratins, CK7, and vimentin, and the negative immunoreaction for EMA, CK20, GATA3, p504S, S100, HMB45 and CD45 certify the renal or urothelial cell origin of the tumour [18, 24, 80]; supplementary, the positive expression for E-cadherin, INI1 and CA9 differentiates a rare variant of RCC, namely the collecting duct carcinoma [80].

The analysis of p53 tumour protein, that functions as a tumour suppressor with role in apoptosis, genomic stability, and anti-angiogenesis, indicates a correlation between its positive expression, poor prognosis and advanced tumour clinicopathological features [24, 36, 65,  81, 82]. However, p53 mutations appear to be rare in RCC and its specificity is low in this context because its positivity does not necessarily translate into a gene mutation [83].

p53 was also studied together with p21, another protein that regulates the cell cycle and apoptosis, in HCV-associated renal tumours [81], in order to explain the relationship with the viral proteins NS5A and NS3 in processes such as the production of reactive oxygen species (ROS) and in the inhibition of apoptosis. This study shows that NS3 viral proteins react with the p53 host protein to form a complex that leads to inhibition of p53 function [81]. HCV NS5A also reacts directly with and co-localizes with the p53 protein in the perinuclear region and inhibits p21WAF1 tumour suppressor transcription in a p53-dependent manner [81].

In BKPyV-, EBV- and HCV-associated malignancies, some authors focus on p16 protein which inhibits the normal cell cycle, in order to evaluate the correlation of this biological marker with classical clinicopathological parameters (tumour stage, tumour grade, disease progression) [24, 36, 79]. The results suggest that p16 could be a potential prognostic marker, useful to predict the progression of these tumours, in a viral and non-viral context, respectively.

Nuclear factor-kappa B (NF-κB) is supposed to be involved in the RCC development, its expression being correlated with the tumour grade. In EBV-associated RCC, the NF-kb nuclear immunopositivity, higher than in non-viral RCC, could sustain the EBV interference in the carcinogenesis mechanism by the activation of NF-κB p65 signaling pathway, leading to the acceleration of tumour progression [36].

In order to evaluate the proliferative tumour potential, two markers, namely proliferating cell nuclear antigen (PCNA) and Ki-67 were studied in HPV- and EBV-associated malignancies, the results revealing their usefulness to estimate the tumours aggressiveness [36, 82].

The relationship with EVB was confirmed only in the sarcomatoid type of RCC [35]. In this aggressive form of RCC, the analysis of the intratumoral lymphocyte infiltrate (TIL) showed that it is predominantly constituted by B lymphocytes that show plasma cell differentiation, in which EBV was located exclusively [35]. This fact was demonstrated by the strongly immunoreactivity for both CD79a and VS38 (markers for B lymphocytes and plasma cells, respectively), and immunonegativity for CD3 (marker for T lymphocytes), CD68 (marker for macrophages) and CD56 (marker for NK cells) [35].

The immune infiltrate was also investigated in a rare case of Kaposi sarcoma developed in a renal allograft ureter, where CD34, together with HHV8, defined the diagnosis [83]. Selected additional markers CD8 (marker for T suppressor lymphocytes), CD19 (marker for B lymphocytes), CD69 (marker for differentiation of regulatory T (Treg) cells) were assessed to monitor reduced immunosuppression by discontinuing MMF and gradually reducing CyA [83].

Reviewer 3 Report

The authors have corrected all flaws in the manuscript. Since the revised manuscript has been sufficiently improved in terms of content and has also gained a lot in terms of readability due to a clearer structure/organization, the submitted manuscript is suitable for publication. In the new text, however, some errors in the English language have crept in, so that a careful English revision in necessary - e.g. as to name only a few:

line 18 and 84: the HPV described in detail in the text (line 348-408) is missing here;

line 268: of "analysed" RCC

line 341: Gonzales "found"

line 344: ccRCC

line 346: nivolumab "warrants"

line 404-5: p53 genes "regardless of HPB type"

line 421: this proves the difficulty in the research

line 433: BKPyV

line 511-3: the predicate is missing

Author Response

Point-by-point response to the reviewers’ and editor’s comments

Title: “The influence of oncogenic viruses in renal carcinogenesis: pros and cons”

We thank the reviewer for giving us the opportunity to improve the quality of our manuscript. We performed all the suggested modification on the clean variant of our manuscript.

Supplementary, we revised tables 1 – 4 and we have corrected all the inaccuracies identified in the first version of the manuscript. Thus, data in tables 1 – 4 are now presented in a more accurate and uniform manner.

Reviewer #3:

Comments and Suggestions for Authors

The authors have corrected all flaws in the manuscript. Since the revised manuscript has been sufficiently improved in terms of content and has also gained a lot in terms of readability due to a clearer structure/organization, the submitted manuscript is suitable for publication. In the new text, however, some errors in the English language have crept in, so that a careful English revision in necessary - e.g. as to name only a few:

  • line 18 and 84: the HPV described in detail in the text (line 348-408) is missing here;

o   Response to Reviewer: We thank the reviewer for this comment. Human Papilloma Virus (HPV) is described in detail at lines 79 – 80:

     The oncogenic role of some viruses in different types of tumours has been well established by the International Agency for Research on Cancer (IARC), the following examples being relevant: Hepatitis B Virus (HBV) and Hepatitis C Virus (HCV) are in-volved in liver carcinogenesis, Epstein Barr Virus (EBV) is known to be responsible for Burkitt lymphoma in parallel with some co-factors, high risk types of Human Papilloma Virus (HPV) are associated with cervical cancer, and oropharyngeal cancers [8].

  • line 268: of "analysed" RCC

o   Response to Reviewer: We thank the reviewer for this comment. We modified the manuscript as suggested (lines 269 - 270):

     In 2002, Wen Y et al., mentioned that 29.6% mRNA of analysed RCC samples were EBV positive by ISH and indirect immunofluorescence staining.  

  • line 341: Gonzales "found"

o   Response to Reviewer: We thank the reviewer for this comment. We modified the manuscript as suggested (lines 340 – 342):

Gonzalez HC et al. found in a prospective study that chronic infection with HCV is associated with having RCC, underlying the relationship between HCV and extrahepat-ic malignancies [55].

  • line 344: ccRCC

o   Response to Reviewer: We thank the reviewer for this comment.  We modified the manuscript as suggested (lines 342 – 346):

In 2020, Tsimafeyeu I et al. evaluated the efficacy and safety of nivolumab in metastatic ccRCC patients with or without chronic HCV infection and the authors found that the administration of nivolumab in warrants further investigation, due to the efficacy and safety profiles observed in this study [59].

  • line 346: nivolumab "warrants"

o   Response to Reviewer: We thank the reviewer for this comment. We modified the manuscript as suggested (lines 342 – 346):

In 2020, Tsimafeyeu I et al. evaluated the efficacy and safety of nivolumab in metastatic ccRCC patients with or without chronic HCV infection and the authors found that the administration of nivolumab in warrants further investigation, due to the efficacy and safety profiles observed in this study [59].

  • line 404-5: p53 genes "regardless of HPB type"

o   Response to Reviewer: We thank the reviewer for this comment. We modified the manuscript as suggested (lines 365 – 367):

In 1996, Pelisson I et al. detected an abnormal distribution of HPV types in skin squamous cell carcinoma from KTRs, and alterations of c-myc, c-Ha-ras, and p53 genes regardless of HPV type.

  • line 421: this proves the difficulty in the research

o   Response to Reviewer: We thank the reviewer for this comment. We modified the manuscript as suggested (lines 384 – 386):

This proves the difficulty in the research of this issue focusing on the possible involvement of viruses in the renal carcinogenic mechanism, and the limitations of current approaches.

  • line 433: BKPyV
  • Response to Reviewer: We thank the reviewer for this comment. We modified the manuscript as suggested (lines 501 – 503):

Therefore, a future research direction could develop a comprehensive and integrated analysis of at least 3 oncogenic viruses (e.g., BKPyV, EBV, HCV), to confirm their role in kidney carcinogenesis. For this purpose, a screening platform for oncogenic virus detection would be useful, based on simultaneously PCR and IHC assays.

  • line 511-3: the predicate is missing

Response to Reviewer: We thank the reviewer for this comment. We modified the manuscript as suggested (lines 462 – 464).

The most frequently used oncolytic viruses include adenovirus, herpes simplex virus, Newcastle disease virus, measles virus, reovirus, and parvovirus. These viruses, which were studied in many research settings, are known that are having a minimal human toxicity as they are destroying only tumoral cells specifically, and they have a large spectrum of anticancer activity.